# A New Method for the Collection of Marine Geomagnetic Information: Survey Application in the Colombian Caribbean

Karem Oviedo Prada [1,2,*], Bismarck Jigena Antelo [1,*], Nathalia Otálora Murillo [2], Jeanette Romero Cózar [1], Francisco Contreras-de-Villar [1] and Juan José Muñoz-Pérez [1,*]

1   Puerto Real Campus, University of Cadiz, 11510 Puerto Real (Cadiz), Spain; jeanette.romero@uca.es (J.R.C.); francisco.contreras@uca.es (F.C.-d.-V.)
2   Oceanographic and Hydrographic Research Centre of the Caribbean, Barrio Bosque, Sector Manzanillo, Escuela Naval de Cadetes "Almirante Padilla", Cartagena de Indias 130001, Colombia; notalora@dimar.mil.co
*   Correspondence: ing.karemoviedo@gmail.com (K.O.P.); bismarck.jigena@gm.uca.es (B.J.A.); juanjose.munoz@uca.es (J.J.M.-P.)

**Abstract:** In recent years, the Oceanographic and Hydrographic Research Center (part of the General Maritime Directorate of Colombia (DIMAR) has made important efforts to advance research in the field of marine geophysics, in particular, the techniques of geomagnetism, sub-bottom profiling, and side-scan sonar, the first being the most developed at the present time. A method is presented for the acquisition of geomagnetic data in marine environments, as used by DIMAR in the Colombian maritime territory. The development of the geomagnetic method not only offers the opportunity to advance basic scientific knowledge, but it is also of great importance in support of national sovereignty issues. Among other applications, the most representative uses of the geomagnetic method are the location of pipelines and metal plates, detection of buried ordnance, identification of sites of archaeological interest, and the identification and characterization of geological structures. As a result of testing the method, a grid of geomagnetic data was surveyed in an area close to the Island of San Andrés in the north-west of the Colombian maritime territory. The survey was prepared with a regional geometric arrangement, the result of which was compared with survey data obtained from the National Oceanic and Atmospheric Administration (NOAA) magnetic data repository and carried out in the same study area. Despite the long time interval between the two surveys, almost 50 years, no significant differences were observed in terms of the analyzed variables. Finally, results show negligible differences between the magnetic data obtained for the years 1970 and 2018 for all the variables measured, such as the inclination, declination, and total magnetic field. These differences may be attributable to a geological component or also to the acquisition and processing methods used in the 1970s.

**Keywords:** marine geophysics; magnetic method; Colombian Caribbean; DIMAR; CIOH

## 1. Introduction

The increase in marine geophysical activity in recent years has provided essential data for evaluating theories about the origin of oceans and continents. Of the different methods used to explore the sea floor and underlying mantle, the magnetic field and its measurements have proven to be one of the most powerful tools for discovering and delineating structural and geological patterns [1].

According to Ewing et al. [2], before World War II, almost all marine magnetic observations had been made by the research ship "Carnegie" (1909–1929), which was specially built to work along widely spaced lines in the Atlantic, Pacific and Indian oceans. After the war, the fluxgate magnetometer, originally developed as an airborne instrument for detecting submarines, was adapted for marine applications by the Lamont Geological Research Observatory. These were the first measurements made with a magnetometer towed by a ship. Later, for work in the maritime field, the fluxgate magnetometer was replaced by

the proton magnetometer, having the advantages of absolute field measurement and not requiring orientation of the head [3].

The use of the geomagnetic method is widely known globally, for its various local and regional applications [4]. Due to its high costs, which involve equipment and logistic development and multiple applications, this geophysical method is generally undertaken by foreign private companies with commercial aims, as the high costs prove to be a disadvantage for state organizations dedicated to science and marine research, which have limited budgetary allocations [5]. Therefore, it can be summarized that established foreign companies, which compose the entire market, dominate geophysical exploration, including those services in limited supply, such as gravimetry and magnetometry and its applications [6–8].

The General Maritime Directorate (DIMAR) is located at Cartagena de Indias, in the Colombian Caribbean Sea. DIMAR started the project "Geomagnetismo Marino" in 2015 with the purpose of recovering research capacity through the use of the G-882 marine magnetometer from geometrics. One of the recovery activities included training on the handling of the magnetic sensor and data acquisition. For the former, a document was produced [9] in which a vast database and manuals were compiled, which served as a base for the production of the following geophysical work methodology.

The need to propose a work methodology was pressing, as there was no record in Colombia of any other public entity carrying out this type of scientific research. Therefore, the efforts of the Caribbean Oceanographic and Hydrographic Research Centre (CIOH) were aimed at the standardization of the guidelines and the parameters required for the optimization of the marine geomagnetic method. After much effort and field tests in the Colombian Caribbean, a methodology has been obtained that offers high-quality marine geomagnetic data collection.

Thus, the following method aims to sequentially show the planning and acquisition of geomagnetic information in deep marine environments in Colombian territory on board the oceanographic research vessel ARC (Navy of the Republic of Colombia) Providence. The guide has become a tool that provides an effective and efficient response for geophysical research at the service of the nation. For this work, a bibliographic compilation was carried out taking into account aspects such as the verification of magnetic sensors and operators that can be powerful sources of magnetic noise. [10]. A fundamental aspect in the survey was to determine the distance at which the magnetometer sensor must be towed to reduce the magnetic effects of the vessel. Finally, the optimal lateral spacing between the lines also had to be considered, which is directly related to the depth of the water [11].

Moreover, geomagnetism is a geophysical prospecting method, applicable to the oil industry, and also mining and archaeological artefact explorations [11–13]. In mineral exploration, magnetometry is widely used to directly prospect for magnetic minerals, such as magnetite and other ferromagnetic minerals, and the method stands out for its speed and low cost. This method is the most widely used in geophysical surveys, at local and regional scales, and it is based on the study of the Earth's magnetic field and its variations, as a consequence of additional magnetic fields produced by magnetized rocky bodies positioned on the surface and close subsoil [14,15].

The magnitude measured in the magnetic method is the Geomagnetic Field, which is related to the magnetization of the environment and which, in the majority of materials, appears when a magnetic field is applied to a body [16]. In the magnetic method, the objective is to investigate the geology of the subsoil, from the variations in this geomagnetic field, resulting from the magnetic properties of the underlying rocks [17,18]. Not all the rock-forming minerals are magnetic, but certain types of rock contain sufficient magnetic minerals to be able to produce significant magnetic anomalies, such as iron and magnetite, among others. The influence of the total magnetic field can be measured anywhere on earth, with a certain direction and intensity, subject to periodic variations and non-periodic disturbances, the magnitude of which on the planet's surface can vary from point to point from 25,000 to 65,000 nT [19,20].

When a magnetic material is placed in a magnetic field, the material is magnetized, and the external field of magnetization is reinforced with the induced magnetic field in the material. This is known as induced magnetization, and it is based on the magnetic susceptibility of the materials, (understood as the degree of magnetization of a material in response to a magnetic field), and the magnitude and direction of the magnetic field [21]. When the external field disappears, the induced magnetization disappears immediately, but some materials retain a residual magnetism, and its direction will be fixed in the direction of the inductive field [22]. The residual magnetism reflects the history of the material. Thus, there is a contrast of magnetism between an anomalous source and the adjacent lateral formations. These two types of magnetization are due to spontaneous magnetization, which is a property of the ferromagnetic minerals in the Earth's crust [23].

To calibrate the data, check its reliability, and study the variation of the new data, it was necessary to have a reference work. To do this, we took into account previous work carried out between 1970 and 1971, obtained from the repository of the National Oceanic and Atmospheric Administration (NOAA). More specifically, from the Marine Geology and Geophysics data from the National Centers for Environmental Information (NCEI), formerly the National Geophysical Data Center or NGDC [24].

The objective of this work is to present a method to carry out magnetic surveys that is compatible with other techniques used in different areas of engineering and science (hydrographic surveys, side-scan sonar, search for magnetized bodies, search for archaeological remains, etc.). To evaluate the quality of the work, the results obtained in a recent campaign were compared with those from previous ones, and the differences were analyzed.

## 2. Materials and Methods

### 2.1. Study Area

The study area is located in the Colombian Caribbean Sea and more specifically to the south of the archipelago of San Andrés, Providencia and Santa Catalina (SAPSC). The geophysical survey was carried out in the area located between San Andrés Island, Cayos de Albuquerque Island and Cayos de Este-Sudeste Island, within the polygon marked in yellow, as can be seen in Figure 1. The study area covers an area of approximately 2040 km$^2$.

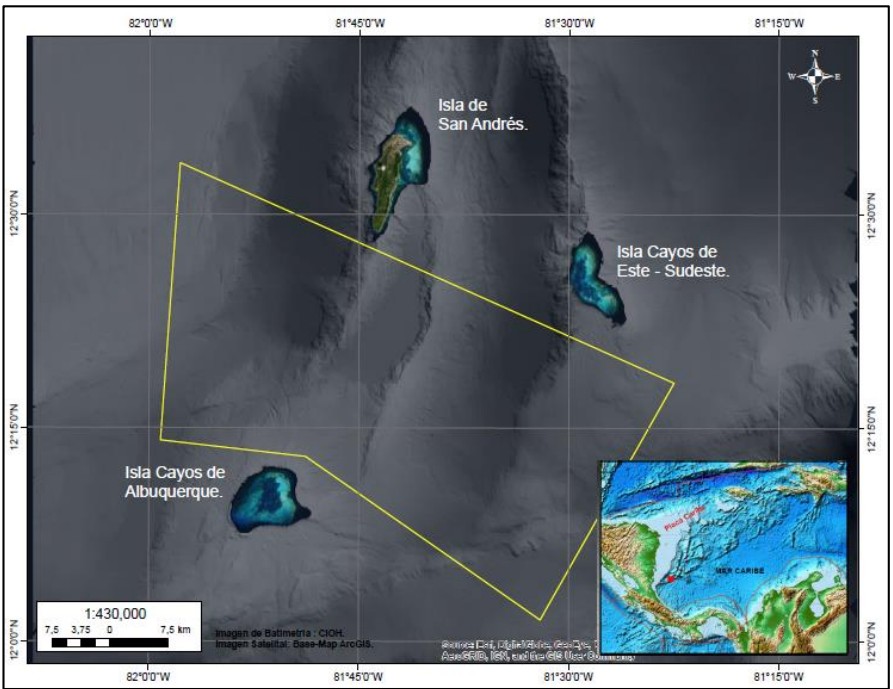

**Figure 1.** Polygon of acquisition in study area.

A magnetic survey measures the local magnetic field characteristics of a certain region. This type of technology only detects minerals and/or materials that respond to magnetic fields. For this reason, its applications are mainly aimed at mineral exploration, but it can also be useful for the exploration of coal, oil, and gas and in the detection of shipwrecks. [2,6,25,26]. A geophysical survey consists of different phases.

### 2.2. PHASE 1. Planning of the Acquisition Campaign

The form of the geophysical survey is established in this phase, and the times, the necessary inputs, and the possible unforeseen events that may occur at sea, are estimated.

Before planning the data acquisition, the study objective and the scale of the work (local or regional) should initially be taken into account. The configuration and length of the lines to be acquired will depend on these.

The generation of the acquisition grid is made based on the sought objectives. It is important to consider whether it is required to determine the regional magnetic field (e.g., changes of magnetic polarity reflected in the marine magnetic anomalies, regional guidelines, etc.), or to determine local geologic anomalies (e.g., geologic bodies and structures), or to identify the anomalies due to metallic objects produced by humans [27]. This is related to the fact that the geometric arrangement of the acquisition must take into account the spatial resolution of the body to be characterized—that is, the smaller the object, the denser and less spaced the survey grid must be.

It is also important to take into account the sensitivity of the sensors, since, to recognize an anomaly, this must be several times greater than the sensitivity (resolution) of the magnetometer and the external noise level. It is important to define this parameter to know if the object is detectable on the surface and, in such a case, how much the readings in the profile, and the distance between adjacent profiles, would have to be spaced (spacing of the grid). Ideally, a grid should be shaped to cover the whole area in such a way that the anomaly can be always detected by a profile. This means that there must be some overlapping between profiles [9].

Additional magnetic information is required, whether from magnetic observatories or from a Base Station near the survey area, with the purpose of improving the quality of the data. In this case, a Geometrics G-862 RBS Base Station (Figure 2) was used, which was acquired by the General Maritime Directorate in 2015. This was positioned at a minimum radius of 60 m from any source of electromagnetic interference. This reduces the errors that can occur in the data, due to fortuitous cases, such as electromagnetic interference from the solar field.

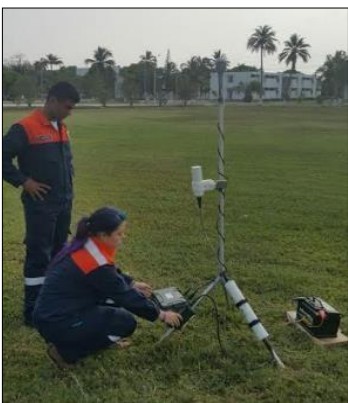

**Figure 2.** Installation of the Geometrics G-862 RBS Base Station.

Starting the planning activity, it is essential to have high-resolution bathymetric data, in order to support the identification of the geological structure that is required to be recorded with magnetometry [28]. As the objective of the project was to determine the magnetic anomalies, generated by the volcanic bodies and geological structures (faults)

located to the south of the SAPSC, in the vicinity of San Andrés Island, Cayos de Alburquerque Island and Cayos de Este- Sudeste Island (Figure 1). The survey lines were carried out, taking into account the geoforms displayed in the bathymetry. For this reason, the lines are established perpendicular to faults or other structures, in regular meshes, where it is ensured that the separation between lines is equal to the estimated minimum distance between the sensor and the magnetic object or target. This is why it is recommended to take into account the depths at which the survey will be carried out [29].

The area included in the geophysical research polygon, in which it was planned to undertake five main lines of acquisition, with a NW-SE direction (azimuth of 300°), with lengths between 70 and 57 km, and a separation of 7 km. The six control lines, oriented perpendicularly to the main lines, are distributed with a spacing of 23.50 km, and they have of an average length of 32 km (Figure 3). In order to calculate the days needed for the survey, the total length of the lines at the optimal survey speed in linear nautical miles was considered, assuming 24 working hours per day [9]. This calculation is shown in Tables 1 and 2.

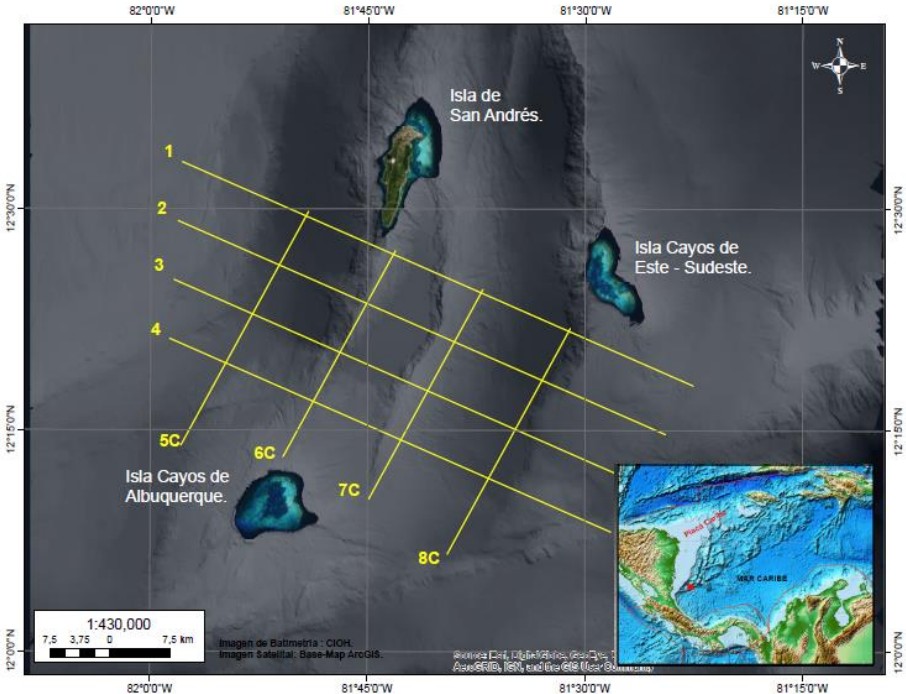

**Figure 3.** Lines of survey.

**Table 1.** Estimation of duration of the survey in days from linear nautical miles.

| Survey | Number of Lines | Meters | LNM | Time in Days |
|---|---|---|---|---|
| San Andrés | 4 | 317,087 | 171.213 | 2 |
| Control lines | 4 | 92,784 | 92.784 | 1 |
| SURVEY | | | | |
| 120 LNM = 24 h | | | | |
| 263.99 LNM = 3 Days of survey | | | | |

Once the configuration of the survey was established, the times for the voyages and duration of the acquisition were estimated. As mentioned previously, it is important to maintain good data density that can adequately represent the objective; that is to say, the optimum survey speed was 5 knots, assuring that the intensity of the signal was stable, and 10 samples per second were obtained.

Finally, a meticulous control was made to ensure that the acquisition of the data was carried out successfully. In the case of consumable equipment, such as RS232-USB converters, it is ideal to have spare parts, in case of unexpected events.

**Table 2.** Estimation of time of the operation, including the displacement.

| Activity | Days | Start Date | End Date |
|---|---|---|---|
| Voyage: Cartagena to San Andrés Island | 02 | D | D+2 |
| Execution of the magnetometry survey in the deep waters off San Andrés Island | 03 | D+3 | D+6 |
| Voyage: Study area to Cartagena | 02 | D+2 | D+8 |
| | | TOTAL DAYS OF OPERATION: | 08 |

### 2.3. PHASE 2. Data Acquisition

The oceanographic research vessels, ARC Malpelo and ARC Providencia, were enabled to operate with the Geometrics G-882 marine magnetometer [30], property of the DIMAR (Figure 4). This apparatus has a broad range of detection for ferrous materials of various sizes and a sensitivity of <0.004 nT/πHz rms, which increases the probability of detection. It has a hydrodynamic design that helps reduce the probability of rock incrustation, and it operates to a depth of approximately 2750 m, and at temperatures from −35 °C to 50 °C. The cesium–vapor sensor is at the rear of the "fish" in the cylinder that forms a T with the longest axis, where the direction of the sensor can be modified; this was vertical, as the work was to be carried out in equatorial latitudes. Finally, the sampling interval ranged from one sample every three seconds, to twenty samples per second, with an absolute precision of <2 nT. The acquisition of field data was carried out with MagLog software from Geometrics Inc.

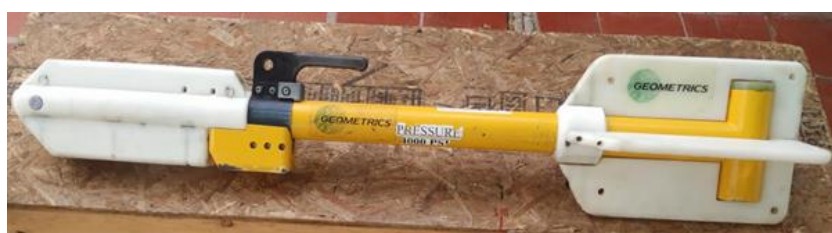

**Figure 4.** Geometrics G-882 marine magnetometer. Source Karem Oviedo Prada, 2020.

### 2.4. PHASE 3. Office and Data Processing

In this office phase, the data were analyzed and filtered and subsequently processed. Oasis Montaj version 8.5 software from Geosof was used for this, and ArcGIS version 10.7 software from ESRI was used for charting.

For this work, the data obtained from the NOAA repository of magnetic surveys carried out in the study area between 1970 and 1971 were used as a reference. A hydrographic and bathymetric survey was performed according to technical specifications of the International Hydrographic Organization (IHO), S-44 publication for Order 2 requirements [31,32]. These regulations guarantee the quality and standardization of the results. This geophysical working method is applicable in all the deep waters of the Colombian marine territory, which are considered to be from the isobath of 100 m, to the maximum registered depth of 4600 m. This specification is also stated by Standard S44 of the IHO [31,32], which recommends that Order 2 surveys are limited to areas deeper than 100 m [33]. Nevertheless, those reference data were about 50 years old, and there were no other modern data available for the study area until the current survey. However, nautical charts 1624 and 004 edited by the CIOH in 1998 nd 2018, respectively [34,35], were taken into account to study the temporal and local variation of the geomagnetic field in the area. Thus, the declination, due to the annual variation effect, was corrected and was 4°18′ (W) in 2020. Likewise,

the magnetic declination was compared with the data published in the AIP COLOMBIA Report of the Gustavo Rojas Pinilla Airport on San Andres Island, for 16 July 2020, which was 02°48′ W [36]. The difference in declinations is due to the separation between the San Andres Airport (North of the island) and the area where the magnetic declination is defined in the nautical chart 004, which is located about 90 km NE of the airport. For the analysis of magnetic declination, the procedure specified by Udias and Mezcua [23] was followed.

In this work, the Minimum Curvature Gridding or Splines method and Geographic coordinate system were used. The Gridding method refers to the process of interpolating data onto an equally spaced grid of "cells" in a specific coordinate system. This interpolation method estimates values using a mathematical function that minimizes the curvature of the surface, resulting in a smooth surface that passes exactly through the input points [37–39].

### 2.5. Components

Going into the field, some indispensable elements must be taken into account to carry out an optimal acquisition. For example, the magnetometry sensor and the portable winch with 300 m of telemetry cable were specifically adapted to collect the geophysical information.

The vessel ARC Providencia (Figure 5) has special adaptations, such as a winch with 2800 m of telemetry cable (Figure 6), a wet laboratory aboard the ship, and the computer center where the magnetic data are visualized and stored in real time.

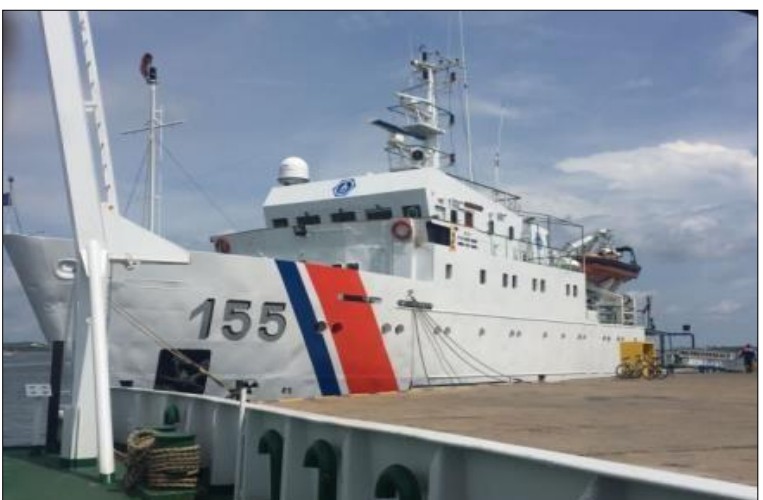

**Figure 5.** Research vessel ARC Providencia.

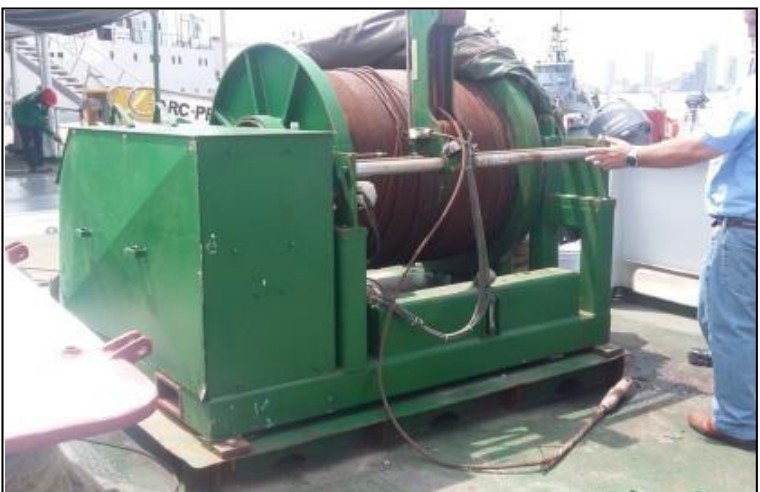

**Figure 6.** 2800 m geophysical research winch on board the ARC Providencia.

The assembly used in the vessels is shown in Figure 7. The magnetic data are communicated from the sensor and submerged in the water, passing through the winch, the on-board cable, and finally, arriving at a "junction box" where the magnetic data are related to those of the positioning obtained by the Global Navigation Satellite System [33,40]. From there, they are transmitted to and visualized in the computer by means of MagLog software [41], as shown in Figure 8.

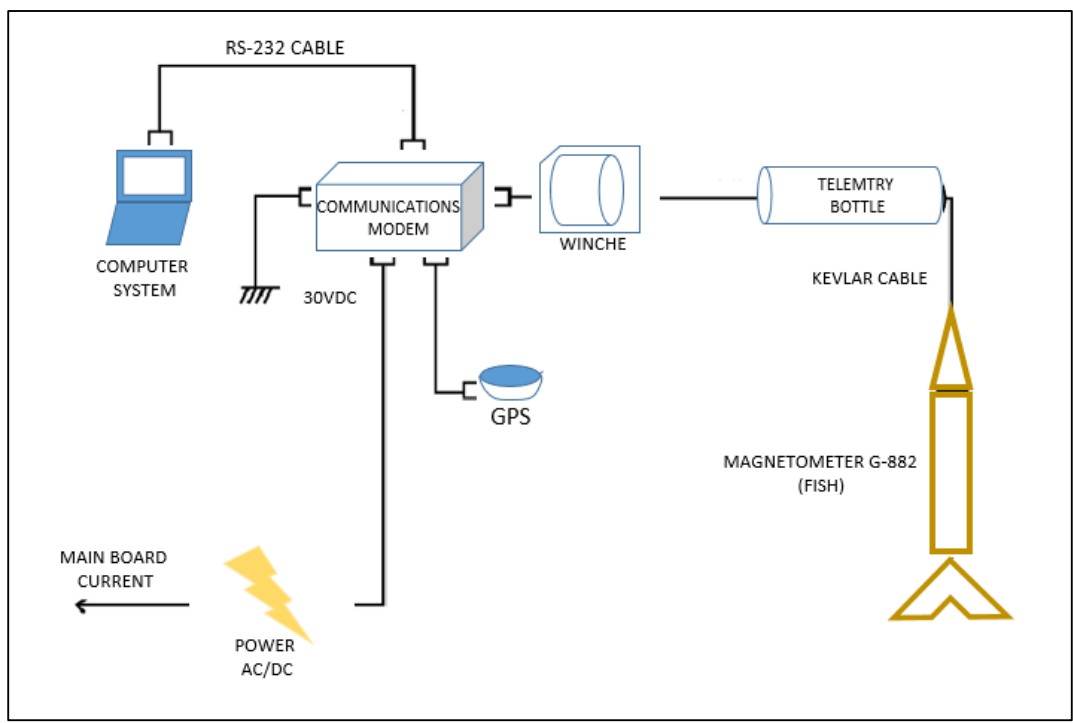

**Figure 7.** Assembly for the magnetic data acquisition.

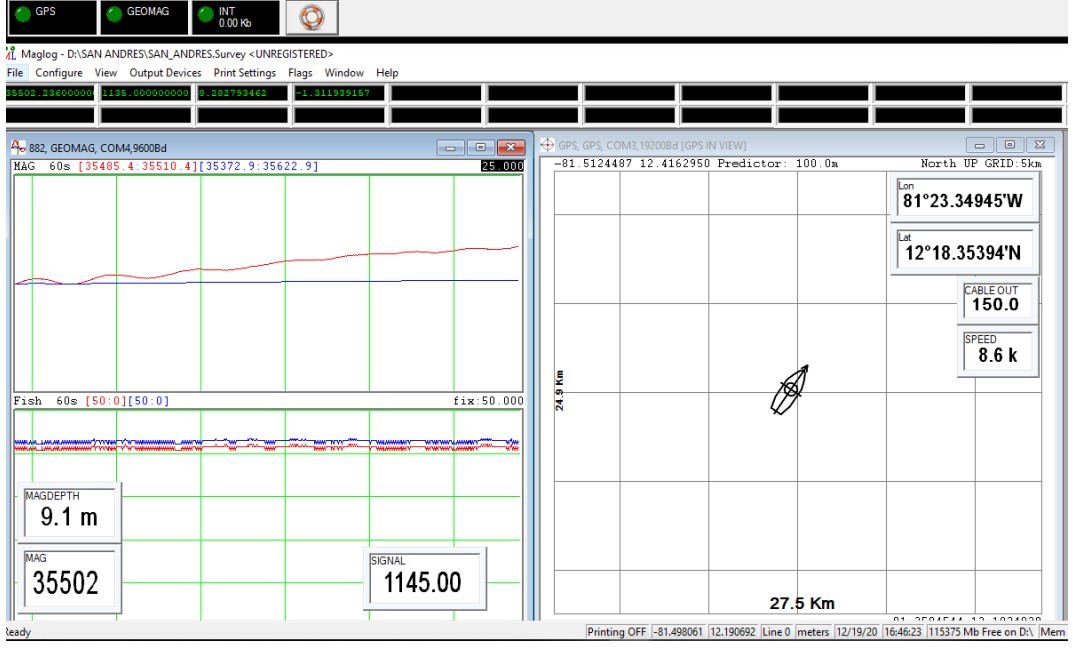

**Figure 8.** Visualization of geomagnetic information in MagLog software in real time.

In Figure 8, showing the visualization of data in real time, the red box to the left is the navigation window, where the position of the vessel and its course are shown. In addition, the lines that the helmsman must follow, according to the planning instructions, are indicated in this window [42]. The blue box, to the right, shows the curves of the data expected, or calculated, by the International Geomagnetic Reference Field (IGRF) model in that precise geographic position and the actual collected data. The small green indicators show the intensity of the signal, the magnetic data, and the positioning. All these indicators must be green, so that the data are correctly acquired. The red rectangle, in the extreme lower right, indicates that it is not recording, and it must be pressed to begin to record the data during the acquisition [37].

The acquisition is made with the help of the ship's personnel, taking into account certain guidelines, such as a maximum velocity of 5 knots, and a separation from the sensor of at least three times the length of the vessel, which in this case was 150 m. It is important that the "fish" is towed from the stern, as indicated in Figure 9.

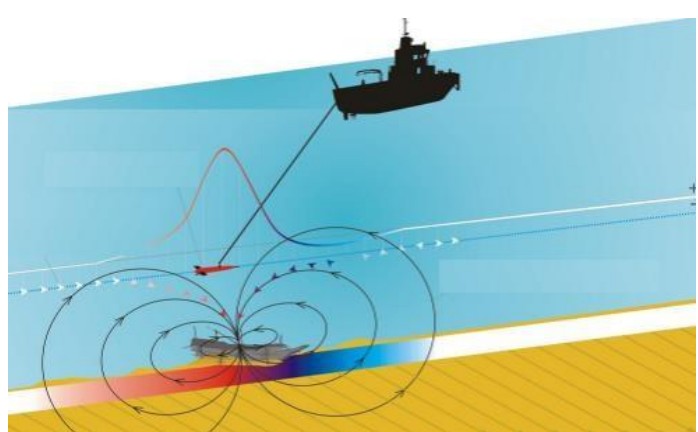

**Figure 9.** System of stern towing of the magnetometry equipment.

Magnetic recording is only useful in straight transects, whereas data from the turns between profile and profile are not considered, as the recorded values are affected by the magnetic field induced by the boat approaching each time a turn is made [43].

The different stages involved in the acquisition of marine geomagnetic data are subjected to a series of decisions that can radically affect the final result of the research [44]. Several usual errors exist that can be committed throughout the process, and which can be classified, according to the development stage of the study. For example, there are frequent errors related to planning that involve poor design of the lines to acquire, which could make it difficult to discern the exact form and size of the anomaly; a measurement is only of interest if the margin of error of that measurement is known. What is interpreted is a collection of data, which is why the sampling must be in accordance with the dimension of the objective to be reached. Other types of errors are associated with the measuring equipment, which can lead to mistaken readings and affect the quality of the data, operator errors, sampling errors, and errors related to environmental noise, among others. Some examples of error handling in geophysical and gravimetric data processing are shown in [45,46].

## 3. Results and Discussion

In the application of the method, some setbacks were presented in terms of what was planned. These were due to logistical issues with the vessel. It is also worth mentioning that the data collection was carried out on board the ship ARC Roncador, with a 300 m portable winch. The geometric arrangement had to be slightly less extensive than originally planned, as shown in Figure 10.

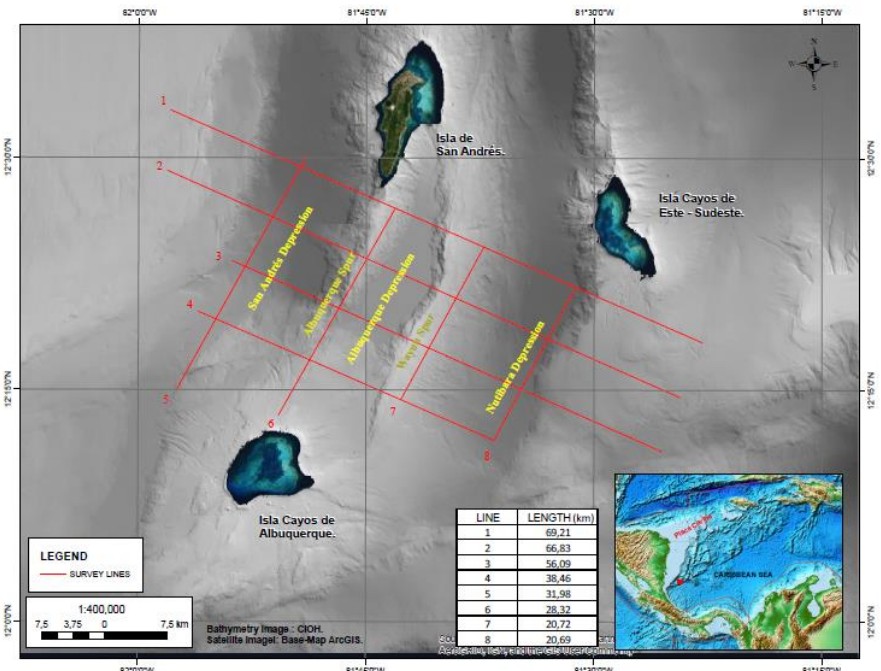

**Figure 10.** Geometric adjustment of the geomagnetic acquisition.

The geophysical study comprises the data collected between 20 June and 1 July 2018, in an area south of San Andrés Island, comprising four lines perpendicular to the general direction of geological structures, with a maximum length of 70.67 km, and four lines parallel to these formations, with a maximum length of 31 km. A grid-shaped geometric arrangement was preserved to provide good resolution for a regional geological study. The general direction of geological structures and geoforms has a northeast direction [47,48].

The chart of the total field of collected data appears in Figure 11, and it shows the magnetic surface of the collected data, after processing for corrections of diurnal variation, delay, direction in degrees, and of the IGRF mode [37]. A significantly positive anomaly was observed in this area, above the Nutibara Depression. The variations were in a range of −170.48 to +159.37 nT.

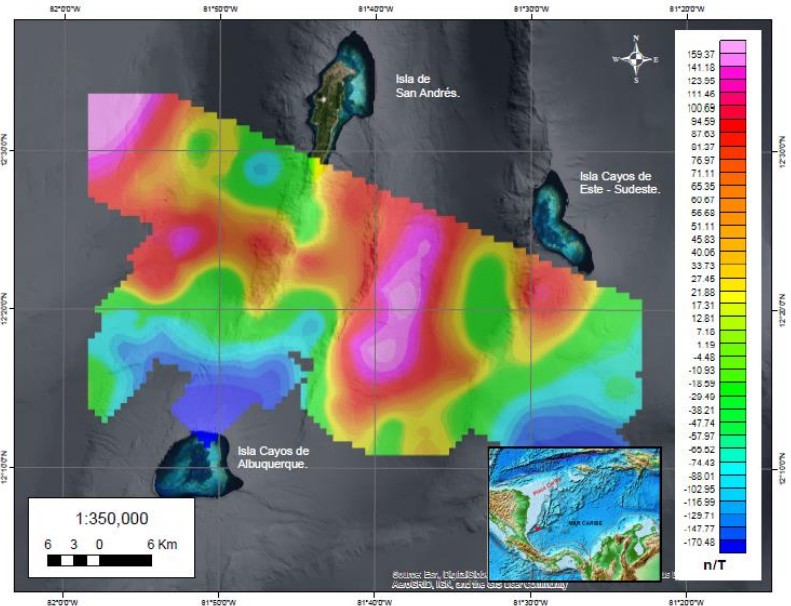

**Figure 11.** Geomagnetic surface of the total field with corrections.

In order to compare the obtained data with pre-existing data, a review was made of the bibliographical material and the data available from possible geomagnetic surveys carried out in the area. As a result, two research cruises were identified, giving free access to marine geomagnetic data from the National Oceanic and Atmospheric Administration [24], which are represented in two survey lines related to geophysical data that contain seismic, side-scan sonar, and magnetometry information. The first downloaded file of the zone, identified by code CH100L12, was collected by the Woods Hole Oceanographic Institution of the United States (WHOI) in 1971. The second file of the zone, identified by the code V2808, was collected by the Lamont–Doherty Earth Observatory of the United States between 1970 and 1971. The two compiled data lines appear in Figure 12.

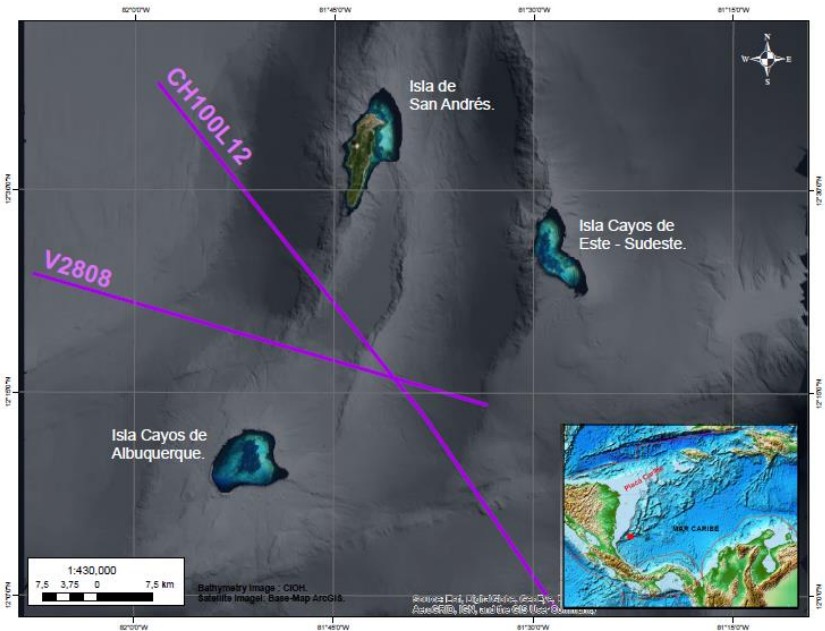

**Figure 12.** Tracking line chart of oceanographic cruises that acquired geomagnetic data in 1970 and 1971.

Taking into account that the magnetic field is dynamic and presents significant annual variations, an attempt was made to find geomagnetic information for the area, finding the only free access data to be those previously described, with the possible source of error of an elapsed time of around 50 years between both surveys. More current geomagnetic data on the study area have been found; however, these are global data obtained through NCEIs (National Center Environmental Information) global aeromagnetic project pertaining to the NOAA. They are aeromagnetic data obtained for the study and modeling of the Earth's magnetic field and have a much higher scale of resolution. In addition, their correction processes are different from the data obtained in situ, and specifically from those obtained in the geomagnetic surveys presented in this work. For those reasons, these aeromagnetic data were discarded for comparison, as they were incompatible in terms of resolution and processing [24]. On the other hand, the variations presented by the magnetic data during the half century between the two surveys have also been taken into account in this work. These variations have an important component of anomalies due to geological sources that have persisted in the study area during that time.

The magnetic information downloaded from NOAA [49] has an extension MGD77T and contains the positions of the tracking line and the data with corrections for diurnal variation and the IGRF. To be viewed on a common surface, the two geodatabases were joined and charted to WGS 1984 UTM Zone 17N, corresponding to the projection of the study area [39,50]. With the positional and magnetic intensity data, a magnetic surface was generated (Figure 13) showing positive and negative anomalies that vary from −328.99 to +40.48 nT.

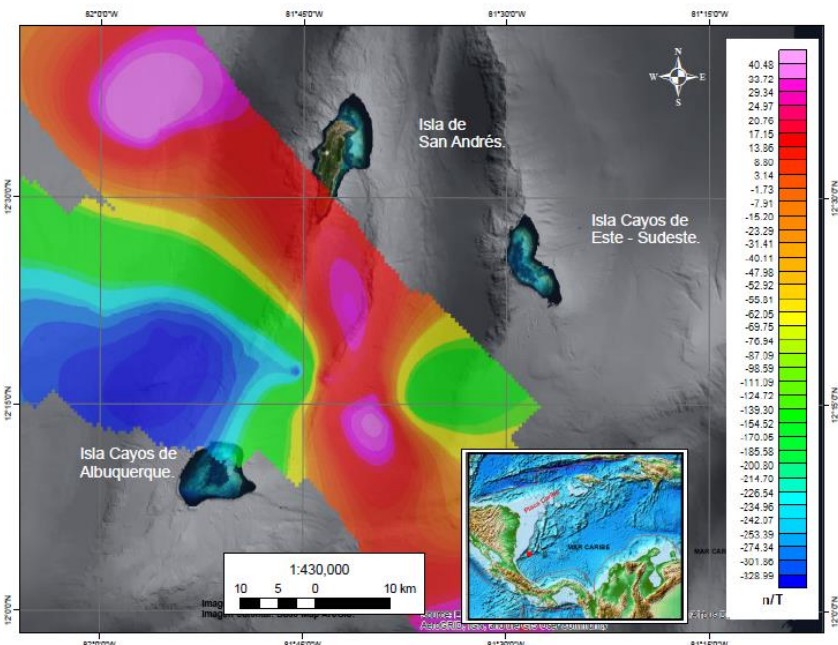

**Figure 13.** Geomagnetic surface of the total field, corresponding to the National Oceanic and Atmospheric Administration (NOAA) data.

In order to visualize the surface of Figure 13, a magnetic grid was generated by the Minimum Curvature Method [38], with a cell size corresponding to 1000. In this image, magenta colors are observed that are associated with positive magnetic peaks, which are located toward the northwest and over the Nutibara Depression and the Wayuu Spur; the geoforms are mentioned in Figure 10.

The layouts of the two magnetic grids are shown in Figure 14, where the positive anomalies are similarly identified on the Wayuu stimulus and the Nutibara depression, to the east and northwest, at the low Nicaraguan elevation. As for the negative anomalies, the magnetic bass located on the areas near the island of Cayos de Albuquerque stands out.

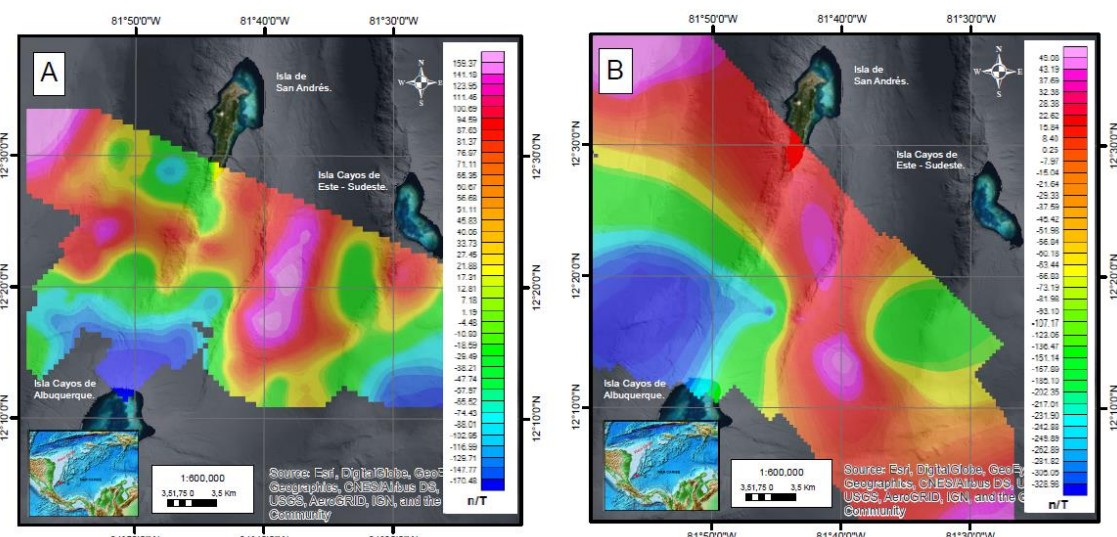

**Figure 14.** Geomagnetic surfaces of the total field. (**A**) Surveyed magnetic field. (**B**) Magnetic field from data downloaded from the NOAA.

It is important to mention that the color scales are not associated exactly with the same ranges on the two surfaces, but they are very close, remember that the color blue is always

associated with low magnetic and pink with high magnetic [51]. These differences are due to the main factor that it is a time difference between the two surveys of around 50 years, and from which other factors that influence the acquired data can be derived, such as the accuracy of the magnetic sensor, the disposition of the field magnetic model in 1970, which presents variations with respect to that of 2018. This due to the displacement of the field and the geometric arrangement used for each case.

Comparatively speaking, in Figure 14A, the magnetic peaks are within the range of $-170.48$ to 159.37 nT, while in Figure 14B, the magnetic lows and highs range from $-328.98$ to 45.08 nT, indicating a variation of 329.85 nT in Figure 14A and 374.06 nT in Figure 14B.

Using the NOAA magnetic calculator [52] and working with the magnetic data observed for the years 1970 and 2018, small but significant differences are identified in all the measurement variables, such as the inclination with $1°46'47''$, the declination with $5°2'41''$, and the magnetic field represented by 4415.7 nT. These results show a variation of the magnetic field across the timeline. It is important to highlight that these anomalies identified in the study area are geological in nature, which infers that they will be present as sources of magnetic anomalies for a very long period of time.

The significant magnetic anomalies, which can be seen in Figure 11, marked in fuchsia/magenta, correspond to a magnetic high. In Figure 10, it can be seen that this anomaly corresponds to the geomorphology of the Nutibara depression, and that it could be generated by some type of mineral deposition that can be found, associated with ferrous materials, or also with volcanic material with a high iron content compared to its geological environment. On the other hand, magnetic lows are observed north of the Cayos de Alburquerque Island, which would seem to be a contradiction, since its morphology is typical of a seamount, and, therefore, its magnetic response should be high [48,53,54].

The results of this work are very important for the scientific community, as this is an area where there is an immense lack of data. The data obtained in the survey, carried out in 2017 by the CIOH, 50 years later, are very important data taken in situ, with a high resolution that make them very reliable and precise, and they are unique in the area of the Archipelago of San Andres, Providencia and Santa Catalina.

We have carried out an exhaustive search for data and geomagnetic surveys in the study area and, although no new works have been found, similar or related works have been carried out in the environment of the study area: magnetic mapping of the northern Caribbean region using marine magnetic data from GEODAS [55], works about gravity and magnetic field referring to hydrocarbon prospects at the Tobago Basin [56], geological description and interpretation in Providencia and Santa Catalina Islands [57], and many others related to tectonics and volcanism [58,59].

## 4. Conclusions

Surveying for the acquisition of geomagnetic data in marine environments is a method that is gaining ground worldwide, and it offers great opportunities for development and advances in new lines of research and scientific knowledge. In addition, it is a technology that supports different research and engineering projects, such as the detection of the location of pipelines and covers, buried ordnance, shipwrecks, identification of sites of archaeological interest, and the characterization of geological structures, among other applications, and also in projects related to national sovereignty and the study of a country's natural resources.

The methodology for marine geomagnetic acquisition has become the prime standard for marine geophysical research for the study of national resources and Colombian sovereignty.

Although geophysical exploration is dominated by established foreign companies, DIMAR now has the capacity to offer geophysical magnetometry services to different countries within its sphere, with excellent technical and human resources, and research equipment and vessels.



After much effort and field tests in the Colombian Caribbean, the geomagnetic acquisition procedure has been standardized as a methodology that can obtain high-quality marine geomagnetic information.

The acquisition of the G-882 marine magnetometer, the application of this methodology to a survey in the Colombian Caribbean, and the development of the magnetometry method have responded to the need of the DIMAR to recover the capacity for scientific research at national level, and scientific leadership in the region, by having an efficient tool in geological and archaeological prospecting supported by two modern, well-equipped scientific research platforms, namely, the ARC Malpelo and ARC Providencia research vessels.

Small differences have been identified between the magnetic data obtained for the years 1970 and 2018, being negligible in the variables measured, such as the inclination, declination, and total magnetic field. These results show a variation of the magnetic field across the very long timeline, so it can be inferred that these anomalies in the study area have an important geological component and will be present for a long time. These differences may also be attributable to the acquisition and processing methods used in the 1970s.

The results of this work are very important for the scientific community, because this is an area where there is a great lack of magnetic data. The data from the survey carried out in 2017 by the CIOH are very important due to the survey resolution reached, having achieved 91,285 data taken in the field at a rate of 20 data/second with an average ship speed of 7 knots, which managed to obtain a datum every 2 cm (0.02 m/datum).

**Author Contributions:** Conceptualisation, B.J.A.; J.J.M.-P. and K.O.P.; investigation and writing—original draft preparation, K.O.P., B.J.A., J.J.M.-P., J.R.C., N.O.M., F.C.-d.-V.; review and editing B.J.A., J.J.M.-P., F.C.-d.-V. All authors have read and agreed to the published version of the manuscript.

**Funding:** The research for the acquisition and capture of geomagnetic data was supported by the Caribbean Oceanographic and Hydrographic Research Center (CIOH), attached to the General Maritime Directorate within the framework of the "Marine Geomagnetism Project". The APC was funded by CIOH and RNM912 Coastal Engineering Research Group of the University of Cadiz.

**Data Availability Statement:** The data presented in this study are available on request from the Centro de Investigaciones Oceanograficas e Hidrograficas de Colombia. The data are not publicly available due to military restrictions.

**Acknowledgments:** This work was possible thanks to the support of the "Marine Geomagnetism" project, financed by the General Maritime Directorate. The authors thank the crew of the ARC Roncador Oceanographic Research Vessel and the staff of the survey area of the Colombian Hydrographic and Oceanographic Research Center (CIOH) for their collaboration during the survey campaigns. We also thank the Captain of the Navy, Germán Augusto Escobar Olaya, Director General of CIOH for his support in the fieldwork and authorisation for the use of CIOH data for the preparation of this document. The authors thank Javier Idárraga García, the editors and the two anonymous reviewers for their comments and suggestions which greatly improved the manuscript.

**Conflicts of Interest:** The authors declare no conflict of interest.

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
