# Peer review of "A New Method for the Collection of Marine Geomagnetic Information: Survey Application in the Colombian Caribbean"

_jmse, doi:10.3390/jmse9010010_

Round 1

Reviewer 1 Report

This is a relatively good article in which there is no significant scientific contribution. The authors made an effort to correct the article according to all the remarks and remove all the shortcomings, thus significantly correcting the text. I think that the article could be interesting for certain scientists who deal with this scientific field and that it could help them in their work.

Author Response

Response 1: Thank you very much to this reviewer. His/her comments have helped us to improve our manuscript.

Reviewer 2 Report

General remarks:

According to JMSE instructions for authors, the abstract should consist of a single paragraph. The abstract lacks the main conclusions and interpretations. In general, the introduction provides sufficient background and includes all relevant references. It can be improved with some specific comments listed below.

The methods description requires supplementation, i.e. provide a source of high-resolution bathymetry.

The results & discussion section lacks interpretation of results, answers for the objectives of the study, comparison with other results in other areas / using similar methodology, and i.e. answer for the question of what your results mean to the scientific community? Did you found any magnetic anomalies?

In the last paragraph of section 3 and the last paragraph of section 4, authors deny themselves writing that there were significant differences and there were small differences between datasets. Please, clarify your point.

Some specific comments:

line 46: it appears that you have an unnecessary comma.

line 54: to be more transparent, consider adding where the DIMAR is located

line 76: It may be unclear who or what This refers to. 

line 131: The comma may be separating the subject and verb in your sentence. Consider removing it. 

lines 210-221: What was the spatial resolution of this old dataset? In lines 158-160 of the introduction, you mentioned that "it is essential to have high-resolution bathymetric data, in order to support the identification of the geological structure that is required to be recorded with magnetometry".

lines 271-275: this part looks like a methodological description of a planned project.

line 276: did you mean transects?

lines 303, 339, 344: the description of used interpolation method should be described in advance in the methodology section

Author Response

Point 1: According to JMSE instructions for authors, the abstract should consist of a single paragraph. The abstract lacks the main conclusions and interpretations. In general, the introduction provides sufficient background and includes all relevant references. It can be improved with some specific comments listed below.

Response 1: We would like to thank the Reviewer for their advice.  The abstract has been merged into a single paragraph and some conclusions have also been added. Furthermore, we have improved the article, answering all the comments sent by the reviewer.

Point 2: The methods description requires supplementation, i.e. provide a source of high-resolution bathymetry.

Response 2: Hydrographic and bathymetric survey was performed according to the technical specifications of the IHO, S-44 publication for Order 2 [40]. These regulations guarantee the quality and standardisation of the results. A new sentence has been added, in lines 210-221, to clarify this topic.

Point 3: The results & discussion section lacks interpretation of results, answers for the objectives of the study, comparison with other results in other areas / using similar methodology, and i.e. answer for the question of what your results mean to the scientific community? Did you found any magnetic anomalies?

Response 3: The results of this work are very important for the scientific community, as this is an area where there is a great lack of data. The data from the survey, carried out in 2017 by the CIOH, are very important, because they are data collected in the area after 50 years, taken at a high resolution, which are very reliable and precise, and currently the only data for the Archipelago of San Andres, Providencia and Santa Catalina areas. In the environment of our study area, other works related to the subject have been carried out, such as magnetic mapping works and gravity and magnetic field studies, geological works, tectonics and volcanism, the details and references of which have been added in a new paragraph at the end of the third section.

Point 4: In the last paragraph of section 3 and the last paragraph of section 4, authors deny themselves writing that there were significant differences and there were small differences between datasets. Please, clarify your point.

Response 4: We would apologise for this error which was due to our poor knowledge of the English language. "quite" has been changed to "small but" at the end of the third section.

Point 5: Some specific comments

Response 5: Responses to specific comments.

line 46: it appears that you have an unnecessary comma.

R: Corrected.

line 54: to be more transparent, consider adding where the DIMAR is located.

R: Corrected.

line 76: It may be unclear who or what This refers to. 

R: Corrected. This sentence has been clarified.

line 131: The comma may be separating the subject and verb in your sentence. Consider removing it. 

R: Thank you for your suggestion, the comma has been removed.

lines 210-221: What was the spatial resolution of this old dataset? In lines 158-160 of the introduction, you mentioned that "it is essential to have high-resolution bathymetric data, in order to support the identification of the geological structure that is required to be recorded with magnetometry".

R: This has been amended in the general remarks, a new sentence has been added to clarify this point.

lines 271-275: this part looks like a methodological description of a planned project.

R: The reviewer is quite correct. This paragraph has been moved to the methodology section.

line 276: did you mean transects?

R: Thank you, the error has been corrected.

lines 303, 339, 344: the description of used interpolation method should be described in advance in the methodology section

R: A new paragraph has been included in the Methodology (lines 232-236) explaining how these calculations were performed.

Point 6. Responses to the Review Report Form

Response 6: The English language and style have been reviewed by a native English speaker. Attached is a signed Letter of Certification.

This manuscript is a resubmission of an earlier submission. The following is a list of the peer review reports and author responses from that submission.

Round 1

Reviewer 1 Report

In abstract it talks about method and then about methodology. These are two separate issues. A methodology includes methods. the article deals with the method and not the methodology.

In the itroduction part, there is no reliable review of the publications of other authors who have conducted similar research. For this reason, it cannot be clearly indicated to what extent the research presented constitutes a new or innovative solution. This part needs to be supplemented.

The publication is a technical note, hence the reviewer does not expect innovation, but a review of the literature on the topic is necessary.

Part: Materials and Methods have been written honestly. It is well written scientifically. The scope of the conducted research should be assessed as large, with their high technical complexity. It determines the positive assessment of this publication.

Part: Results and Discussion section contains a reluctant description of the results and their evaluation. However, it should be supplemented with a discussion of the results with analogous research by other authors.

The bibliography requires considerable expansion.

After the supplements indicated in the review, the article will be suitable for publication.

Reviewer 2 Report

Article "Methodology for the collection of marine geomagnetic information. Survey application in the Colombian Caribbean" needs extensive revision to be acceptable for publication in a journal. 

Individual remarks:

The abstract should be rewritten in accordance with the text of the paper.

The introduction is too long and does not provide a description and overview of previous work on the collection of geomagnetic information, but describes the institution that performs the survey.

The section on materials and methods, like the introduction, is unnecessarily long and it mainly describes magnetic survey instruments. The methods are not described nor accuracy of the resulting survey.

The section results and discussion do not match the title and begin with the area of research, which has no place there. The authors did not comment well enough on the results. The results are compared to a fifty-year-old NOAA model, for which they did not state how it originated. The authors should compare their results with the newer model because geomagnetic data have significant annual variations, and this is a half-century interval.